# Optical Sensing Technologies to Elucidate the Interplay between Plant and Microbes

**DOI:** 10.3390/mi14010195

**Published:** 2023-01-12

**Authors:** Asia Neelam, Shawana Tabassum

**Affiliations:** Department of Electrical Engineering, The University of Texas at Tyler, Tyler, TX 75799, USA

**Keywords:** biosensor, localized surface plasmon resonance, fluorescence, lux biosensors, rhizosphere, plants, phytohormones, exudates, microbial community, biotic and abiotic stressors

## Abstract

Plant-microbe interactions are critical for ecosystem functioning and driving rhizosphere processes. To fully understand the communication pathways between plants and rhizosphere microbes, it is crucial to measure the numerous processes that occur in the plant and the rhizosphere. The present review first provides an overview of how plants interact with their surrounding microbial communities, and in turn, are affected by them. Next, different optical biosensing technologies that elucidate the plant-microbe interactions and provide pathogenic detection are summarized. Currently, most of the biosensors used for detecting plant parameters or microbial communities in soil are centered around genetically encoded optical and electrochemical biosensors that are often not suitable for field applications. Such sensors require substantial effort and cost to develop and have their limitations. With a particular focus on the detection of root exudates and phytohormones under biotic and abiotic stress conditions, novel low-cost and in-situ biosensors must become available to plant scientists.

## 1. Introduction

Microbial diversity exists in a natural soil environment with up to 10^10^ microbial cells along with tens of thousands of bacteria and archaea living under the surface of each gram of soil [1]. Some of these microbes, such as mycorrhizal fungi and nitrogen-fixing symbiotic bacteria substantially contribute to plant nutrition through recycling and utilizing the soil organic carbon as a source of energy [2], fertilizing crops by providing nutrients, controlling or inhibiting plant pathogens, enhancing soil structure by forming microaggregates, mineralizing the organic pollutants in soil, and reducing the reliance on chemical fertilizers to achieve high productivity [3,4]. These free-living soil bacteria are collectively called plant growth-promoting rhizobacteria (PGPR) that invade the root system and facilitate plant development by several mechanisms [5,6]. They are also known as plant health-promoting rhizobacteria (PHPR) or nodule-promoting rhizobacteria (NPR) that stimulate plant-microbe interactions in the rhizosphere, a crucial soil ecosystem habitat [5]. The biogeochemical processes in the rhizosphere influence the activity and composition of the plant’s microbial community. Increasing evidence suggests that bacteria such as Pseudomonas, Azospirillum, Azotobacter, Klebsiella, Enterobacter, Alcaligenes, Arthrobacter, Burkholderia, Bacillus, and Serratia promote plant growth and thus serve as PGPR [7].

Plant-microbe interactions are mutual. The rhizosphere, a complex zone surrounding plant roots, is influenced by root secretions, plant species and their developmental stages, soil properties, nutrient status, land use, and climatic conditions [8]. Studies suggest that the host plant’s unique composition of root exudates plays a major role in determining how the plant’s microbiome is structured, indicating the selective effect of a host plant on plant-microbe interactions [1,9]. These exudates account for 5–21% of total photosynthetically fixed carbon-containing signaling and chemoattractant molecules. These molecules help in recruiting beneficial microorganisms that contribute to pathogen resistance, water retention, and the synthesis of growth-promoting hormones that influence plant phenotype [10]. There are also harmful microbes, termed pathogens, which cause damage to the host plant. Plants recognize and respond to these pathogenic infections via the expression of specific defense or signaling molecules called phytohormones [11]. A better understanding of rhizosphere microbiota and plant health would help manipulate the soil microbiome directly by incorporating specific microbes in the soil or indirectly by modifying management practices to improve crop performance [8]. Moreover, understanding the dynamics and crosstalk between the hormonal signaling pathways would elucidate the defense mechanisms in plants [12].

An analysis of the host plant together with its associated microbiome, typically called holobiont, suggests the coevolution of plants and microbes [13]. Many modern technologies such as next-generation sequencing (NGS) [14], computational tools, omics approaches (metagenomics, transcriptomics, proteomics, metabolomics), and Clustered Regularly Interspaced Short Palindromic Repeats (CRISPR)-based tools have revealed promise for understanding the molecular aspects of plant-microbe interactions, which underlie sustainable agricultural practices [15]. As an analytical tool, biosensors have been extensively used to detect multifarious target substrates in the last few decades. These biosensors convert the chemical interactions into a measurable optical, electrical, or acoustic response [16]. A biosensor is expected to detect target molecules with a high signal-to-noise ratio (SNR), provide high spatial and temporal resolution at the cellular/molecular level, respond quickly, and work under varying environmental conditions such as changes in temperature, pH, or redox states. In addition, the detection procedure should not interfere with cellular processes, cause cellular damage or incur any toxicity [17]. There is a wide variety of biosensors reported in the literature [18]. Over the past ten years, many studies have been published on plant biosensors, demonstrating the progress of this technology and its significance in plant research. Our group is pursuing wearable, flexible, and wireless sensor design for in situ monitoring of phytohormone signaling and dynamics in plants [19,20,21,22,23].

This article is organized into several sections. The present section provides an introduction while Section 2 explains the interplay between plant and microbial communities and how these interactions are mediated. Section 3 discusses the current state of the optical biosensing methodologies that detect pathogen- or microbe-associated molecular patterns (PAMPs or MAMPs) such as cell surface proteins, liposaccharides, DNA, RNA, toxins, etc., or biochemicals (such as root exudates) that mediate plant-microbe interactions. Finally, Section 4 presents the conclusion and prospects of advancing the fundamental knowledge of microbiome dynamics.

## 2. The Interplay between Plants and Microbial Communities

Plants can host a wide range of microbes, collectively known as the plant microbiome, in the rhizosphere (i.e., the region of soil in the vicinity of plant roots), endosphere (i.e., plant internal tissues), and phyllosphere (i.e., stem, leaves, or flowers) as illustrated in Figure 1 [24]. These microbiomes form long-lasting interactions with the host plant, leading to positive, neutral, or negative impacts on crop performance and microbe-mediated biogeochemical processes [25]. The microbial community that is beneficial to the host plant’s health, function, and evolution, are termed microsymbionts for forming a symbiotic relationship with the plant. The other type of microbe functions as plant pathogens, causing damage to the host plant. In most cases, the beneficial effects of microorganisms on plants are not caused by a single microbe, but rather by a consortium of different microorganisms that induce systemic resistance and promote plant growth [26]. Berendsen et al., reported the prevalence of three bacterial genera, *Microbacterium*, *Stenotrophomonas*, and *Xanthomonas*, in the rhizosphere of *Arabidopsis thaliana* when the foliar defense was activated by the downy mildew pathogen *Hyaloperonospora arabidopsidis* [27]. This study revealed that the plant recruited these bacterial species in the root zone to induce systemic resistance against downy mildew. Moreover, the formation of this symbiotic relationship in the primary population of downy mildew-infected plants resulted in a higher chance of survival of the second population of plants grown in the same soil. Microbiomes found in the endosphere and rhizosphere regions have also been shown to suppress plant diseases caused by fungal pathogens *Gaeumannomyces graminis* and *Rhizoctonia solani* (a soil-born pathogen) [28,29]. Similarly, other studies suggested the impact of a consortium of endophytes, including the fungi *Rhodotorula graminis*, and the bacteria *Burkholderia vietnamiensis*, *Rahnella* sp., *Burkholderia* sp., *Acinetobacter calcoaceticus*, *Sphingomonas yanoikuyae*, *Pseudomonas* sp., *Rhizobium tropici*, and *Curtobacterium* sp. on the enhanced drought stress tolerance in poplar plants [30]. In another study, pepper plants inoculated with desert-adapted bacteria displayed higher tolerance to water shortage compared with control plants. The bacteria enhanced the root biomass and length of plants (by 40%), which in turn improved the plant’s ability to uptake water and survive under water stress conditions [31]. Furthermore, mutualism between plants and microbes increases nutrient availability for plants. Beneficial interactions of the host plant with the microbial community contribute to the co-existence of multiple plant species, thereby enhancing plant and microbial diversity. The heterospecific plant-soil feedback responses play an important role in the co-existence of species, ecological succession, and species invasiveness. A meta-analysis conducted by Kutakova et al., suggest that plants grew better in soil conditioned by their closely related species than in soil conditioned by less frequently co-occurring species [32].

However, microbes that act as plant pathogens can directly infect the seedlings and suppress beneficial interactions. There are three main categories of pathogens: *biotrophs,* that feed on nutrients while keeping the host plant alive; *necrotrophs,* that suppress and destabilize the host’s immune system by producing tissue-degrading toxins and enzymes and feed on the dead tissue; and *hemibiotrophs,* that initially behave like *biotrophs* but the transition to *necrotrophs* in later stages of the disease [11]. For instance, *Pseudomonas syringae* (*P. syringae*) strains secrete the effector molecule, AvrPto1, which suppresses immune-related proteins in tomato plants. These effector molecules are specific to the pathogen [33].

The plant produces signaling hormones or phytohormones that include salicylic acid (SA), jasmonic acid (JA), ethylene (ET), abscisic acid (ABA), and auxins (such as indole-3-acetic acid (IAA)). The phytohormone signals are generated in the infected tissue and then circulated throughout the plant via the xylem and phloem. A progressive variation in the phytohormone levels serves as an early signal of plant stress [34,35,36]. Figure 2 shows a simplified model of the phytohormone dynamics in response to a stress condition [37]. SA, JA, ABA, and IAA are among the most important regulators of induced defense mechanisms [33,36,38,39,40,41,42,43]. Progressive variations in their levels have been reported in response to abiotic stressors including drought, salt, and cold/heat conditions [43,44,45,46] as well as biotic stressors including pathogen infection [47,48]. Exogenous application of these hormones is found to mitigate oxidative stress in plants [46,49,50,51,52,53,54,55]. Oxidative stress occurs due to a burst of reactive oxygen species, which are triggered by biotic (attack by microbes, pests, herbivores) and abiotic (drought/floods, temperature variations, soil nutrient/salinity/pH deficiencies) stresses. Root exudates are another form of signaling molecules that mediate the communication of plants with the rhizosphere. Root exudates are primarily composed of sugars, amino acids, organic acids, and vitamins, serving as a rich source of nutrients for the microbial community [56]. These exudates serve as carbon and energy sources for microorganisms living in the rhizosphere, while also profoundly influencing the composition and diversity of the microbial community [57]. Plants release the majority of photosynthates (the products of photosynthesis) into the rhizosphere through roots. Plants also release 100 teragrams (Tg) of methanol and 530 teragrams (Tg) of isoprene each year [58,59]. The interconnected signaling pathways of the compounds secreted from plants are central to the plant’s ability to fine-tune the rhizosphere’s microbiome structure or the induction of defenses in response to stressors. This section describes the various signaling molecules that mediate plant-microbial interactions.

### 2.1. Phytohormone Mediated Plant-Microbe Mutualism

Plant immunity depends on a complex network of phytohormones, chemical signals, and antimicrobial compounds to facilitate defense against environmental stress conditions [60]. These compounds coordinate with each other to regulate plant development, including reproduction, leaf senescence, and response to biotic and abiotic stresses. One of the major auxins found in plants is indole-3-acetic acid (IAA), which regulates root development, cell division, and apical dominance [61,62]. Despite the well-established role of IAA in promoting plant growth, numerous studies demonstrate that IAA plays multifaceted roles during plant-pathogen interactions, including promoting epiphytic colonization, inhibiting host plant’s defense mechanisms, and stimulating alterations in host physiology, which make the host more hospitable to pathogens [39]. In addition, IAA regulates virulence gene expression in many pathogens, promotes the survival of microbes under stress conditions, and acts as a microbial signal for interacting with other microbes in the environment. Various plant-associated terrestrial bacteria produce IAA [63,64,65]. Depending on how the microbe-synthesized IAA affects the plant’s endogenous IAA pool and how the plant responds to this exogenous IAA, the microbes can function as pathogens, symbionts, or growth promoters. Some studies show that IAA exuded by the microorganisms decreases the stiffness of plant cell walls, resulting in the leakage of sugars and nutrients beneficial to the microorganisms [66,67]. The increased availability of root exudate is amenable to bacterial colonization [67]. In many cases, the pathogen produces auxins as virulence factors, while in other cases, the pathogen stimulates auxin signaling in the host. For example, *Pseudomonas savastanoi* and *Pseudomonas agglomerans* pathogens produce IAA to promote gall formation at the infection site of the host plant [39]. *Pseudomonas syringae* produces IAA analogs that interfere with jasmonate and ethylene signaling in plants, resulting in stomata opening and pathogen penetration [39,68]. In contrast, plant cells synthesize IAA upon infection by a tumorigenic pathogen such as *Agrobacterium tumefaciens* [39]. This pathogen genetically transforms plant cells by delivering the T-DNA into the host cell nucleus. This process results in elevated levels of IAA and cytokinin at the infection site, which leads to uncontrolled plant cell proliferation and subsequently gall formation. Moreover, IAA modulates virulence gene expression in several pathogens, including *Dickeya didantii* (that causes soft rot and other diseases), *Agrobacterium tumefaciens*, *Pseudomonas savastanoi*, and *Pseudomonas syringae* [39].

The secondary metabolites salicylic acid (SA) and jasmonic acid (JA) play a central role in plant’s defense against pathogens [69]. Particularly, the SA signaling in plants is associated with resistance to biotrophic microbial pathogens, while JA-mediated responses are linked with necrotrophic pathogen attack [47,48]. For example, the SA pathway mediates the resistance against the biotrophic pathogen *Pseudomonas syringae*, whereas JA is effective in mediating resistance against *Botrytis cinerea*. Upon exposure to pathogens, the reactive oxygen species in plants are up-regulated to trigger the expression of genes that encode antimicrobial or other defense proteins [70]. According to Ross (1960s), systemic acquired resistance (SAR) developed in plants, and is associated with elevated levels of SA resulting from the coordinated activation of several pathogenesis-related (PR) genes, which are likely to make antimicrobial PR proteins [71,72]. SAR represents a long-lasting and enhanced immunity against future infections developed by plants upon exposure to a pathogen. SA and JA signaling pathways generally antagonize each other, which implies that an increased defense response against necrotrophs is followed by an increased susceptibility to biotrophs and vice versa [42,47]. Several plants and a large number of bacteria, such as *Pseudomonas*, *Bacillus*, *Azospirillum*, *Salmonella*, *Achromobacter*, *Vibrio*, and *Yersinia*, have also been reported to synthesize salicylates through non-ribosomal peptides (NRPS) or Polyketides (PKs) biosynthetic gene clusters [73]. Jasmonates released by plants stimulate the interactions between plant roots and beneficial bacteria or fungi. Such interactions alter gene expression to slow down growth and redirect metabolism so that defense molecules are produced and damage is repaired [74]. Likewise, the volatile phytohormone ethylene acts as an important regulator of induced systemic resistance (ISR). Its production triggers the plant’s defense mechanisms such as activation of the phenylpropanoid pathway, phytoalexins, pathogenesis-related proteins (PR), and changes in cell wall structure [75]. Moreover, ethylene stimulates the development of necrosis and inhibits the proliferation of several biotrophs by activating the hypersensitive response (HR). The study by Thomma et al. [76] shows the resistance of the Arabidopsis plant to *Botrytis cinerea* by the activation of a JA-/ethylene-mediated pathway. Ethylene production by microbes may have profound impacts on plant physiology and life history as it accelerates fruit ripening in plants. However, the beneficial microbiota can also control ethylene levels to maintain homeostasis in plants [77]. Other plant hormones such as auxins, gibberellic acid (GA), abscisic acid (ABA), brassinosteroids (BR), oxylipins, and cytokinins (CK) also optimize resistance against invading organisms [69,78,79]. Root-associated microbes such as endophytic or symbiotic microbes also produce phytohormones that induce a defense response against abiotic stresses such as salinity, heat, drought, stress, and metal toxicity or biotic stresses such as pathogenic fungi or bacteria invasion [80]. Table 1 outlines the phytohormones exuded by different microbial strains. Besides the phytohormones, plants also release volatile organic compounds (VOCs) such as monoterpenoids, sesquiterpenoids, and homoterpenoids from leaves, flowers, roots, and other tissues, in response to pathogen attacks [81,82,83,84,85]. The emission rate of plant VOCs is a strong function of stressor agents [86,87]. These VOCs regulate plant physiology by interacting with microbes [88] deterring pests [85] and inducing defense responses [89]. Therefore, the hormonal network collectively plays a pivotal role during any plant-pathogen interactions. 

Guided by the evidence in the literature, in-situ and real-time monitoring can provide new insights into the largely interconnected signaling processes that govern plant-pathogen interactions, as well as uncover new hormones and their roles.

### 2.2. Role of Microorganisms in Plant Nutrition

The rhizosphere zone around plants is teeming with diverse groups of prokaryotes and eukaryotes. In a teaspoon of soil, there are about 8 to 15 tons of bacteria, fungi, protozoa, nematodes, earthworms, and arthropods. Some of these organisms, such as mycorrhizal fungi and nitrogen-fixing bacteria, play a vital role in plant nutrition [2]. Nitrogen-fixing bacteria belonging to genera such as *Bacillus*, Azotobacter, and *Rhizobia*, and fungi belonging to phyla Glomeromycota, Basidiomycota, and Ascomycota are found to be the most beneficial for improving plant productivity [93,94,95,96,97]. These fungi and bacteria can serve as an alternative to chemical fertilizers in sustainable farming practices. The studies of root microbiomes show that the rhizosphere is a land of ecological riches and the plant root systems support a wide variety of microbial taxa [98] that exhibit a diverse array of interactions including competition, commensalism, or mutualism. Many nutrients such as nitrogen, phosphorus, and sulfur (N, P, and S) are bound in organic molecules in natural ecosystems and are thus barely bioavailable to plants [2]. To make these nutrients available to plants, soil microbes such as bacteria and fungi depolymerize and mineralize the organic forms of N, P, and S. Through this process, inorganic N, P, and S are liberated into the soil in the form of ionic species such as nitrate, phosphate, and sulfate, respectively, which are ideal nutrients for plants [99]. Numerous types of bacteria have been identified that aid in the mobilization of nutrients and facilitate the growth of plants. These free-living rhizobacteria are members of the genera *Azospirillum*, *Azotobacter*, *Gluconoacetobacter Beijerinckia*, *Bacillus*, *Paenibacillus*, *Pseudomonas*, *Burkholderia*, and *Herbaspirillum* [100]. Several species of bacteria from the genera Azospirillum, Azoarcus, and Azotobacter fix nitrogen in legumes, sugarcane, and rice plants. However, species of *Burkholderia*, *Azoarcus*, and *Bacillus* enter the roots of rice plants to increase nitrogen uptake by the plants [101,102,103,104]. *Gluconacetobacter*, Herbaspirillum, and Azospirillum, which are common sugarcane endophytes, also contribute to the nutritional well-being of plants [105].

### 2.3. Exudate Mediated Plant-Microbe Mutualism

Root exudates function as a signaling messenger between soil microbes and plant roots, leading to increased microbial activity surrounding roots and “microbial mining” of nutrients [106]. Plants have evolved recognition mechanisms to distinguish between beneficial microbes and those that should be repelled. Thus, the specific molecules such as flavonoids, terpenoids, or strigolactones present in the root exudate could be the targets for plant breeding strategies to engineer the rhizosphere microbiome [2]. Various studies have demonstrated that plant root exudates shape microbial communities in the rhizosphere, or specifically attract mutually beneficial interaction partners [107]. The interaction can vary from pathogenic to symbiotic or beneficial to deleterious depending on the environmental conditions. Based on soil nitrogen levels, rhizobia and nitrogen-fixing bacteria might have a symbiotic or neutral relationship with plants. Under nitrogen-limiting conditions, legumes secrete more flavones and flavanols to attract and initiate legume–rhizobia symbiosis [108]. Rhizophagy refers to the plant’s ability to farm the beneficial microbes to acquire nutrients. Root-associated bacteria are frequently motile and may move away from the plant root to obtain soil nutrients and then return to the host plant to obtain extra carbon and other nutrients [109]. In the process of nutrient transportation, microorganisms carrying nutrients from the soil enter the root’s apical meristem cells. The microbes convert into wall-less protoplasts in the periplasmic spaces of root cells [110]. During root cell maturation, microbes are subjected to reactive oxygen (superoxide) produced by NADPH oxidases (NOX) on the plasma membranes of the root cells. When the reactive oxygen degrades some of the microbes inside the root cells, the microbes release electrolytes to provide nutrients to other cells. In the root epidermal cells, bacteria cause root hairs to grow, which helps bacteria to exit from the tips of the roots, reforming cell walls and changing cell shapes as microbes enter the rhizosphere to obtain nutrients. Micronutrients that are essential for the plants such as organic nitrogen and phosphates are solubilized in the rhizosphere due to the activity of the symbiotic organism and hence can be easily transported into the roots [111]. However, in certain conditions, bacteria secrete “siderophores” that have a high binding affinity for iron to efficiently scavenge iron from their environment [112]. Despite the large body of research studies on the rhizophagy cycle, it is still unknown which nutrients are transferred via rhizophagy or the importance of this process in nutrient acquisition [106].

### 2.4. Influence of Environment on Plant-Microbe Interactions

There exists a complex network of interactions among plants, colonizing microorganisms, and their environment, impacting plant growth and yield [113]. Environmental factors such as soil pH, biogeochemical properties, water availability, salinity, and temperature profoundly impact microbial colonization [114]. The well-known “disease triangle” states that environmental conditions conducive to pathogen virulence and plant susceptibility drive the occurrence of a plant disease [115]. Hence, it is crucial to understand the influence of environmental conditions on plant-microbe interactions to predict disease outbreaks, engineer synthetic microbial communities, and breed stress- and disease-resilient crops. Some examples of environment-induced changes in plant-microbe interactions are provided below.

Plants have evolved immunity against pathogens via two types of defense mechanisms. Plants may recognize the pathogen- or microbe-associated molecular patterns (PAMPs or MAMPs) that include cell surface proteins, liposaccharides, DNA, RNA, and toxins from the invading bacteria or fungi. Recognition of PAMPs or MAMPs triggers the plant’s immune system, thereby working as the plant’s first line of defense against the microorganisms. This defense response is called PAMP-triggered immunity (PTI). However, the pathogens have evolved to possess virulence-effector proteins that enter the plant cells and suppress the PTI. Plants have developed a sturdier immunity called effector-triggered immunity (ETI) to counter pathogen virulence [116]. Studies show that elevated temperature dampens ETI in plants, thus making the plants highly susceptible to virulence pathogens [117]. Although the pathways underlying ETI inhibition by elevated temperature are not fully elucidated, some research studies show that temperature changes result in differential nucleo-cytoplasmic localization of NLR proteins in plants [118,119,120,121]. Besides affecting ETI, the increased temperature reduces the gene expression associated with the plant’s primary defense hormone salicylic acid, leading to increased disease development [122]. On the other hand, plant-microbe interactions can help both the host and microbes cope with temperature changes. For instance, the symbiotic relationship between the tropical panic grass *Dichanthelium lanuginosum* and the fungus *Curvularia protuberate* enabled both organisms to grow at high soil temperatures. However, when they were grown separately, neither the plant nor the fungus could survive at the elevated temperature [123]. Another example is the bacterium *Burkholderia phytofirmans* that makes the host plant resilient to multiple stressors, such as improving tolerance to cold in grapevine, heat in tomato, drought in wheat, and salt and freezing in Arabidopsis [124,125].

Soil water content impacts different aspects of plant physiology and microbe biology. Plants produce an elevated level of abscisic acid (ABA) in response to drought stress. ABA increase in turn triggers a signaling cascade, resulting in physiological changes including stomata closure to reduce transpiration [126]. ABA-induced stomatal closure reduces bacterial entry through stomata [68]. However, ABA signaling interacts antagonistically with the salicylic acid (SA) pathway, inhibiting SA pathyway and thereby compromising post-invasion resistance [127]. In addition, water deficiency significantly reduces the diversity of bacterial community in the rhizosphere and the root endosphere. Drought stress also alters the composition of root metabolites, which eventually may reconfigure the root microbiome. Deciphering these molecular pathways with appropriate biosensors would allow engineering the root microbiome for breeding enhance drought-tolerant cultivars.

Soil nutrient availability significantly impacts plant-microbe interactions. Low nitrogen (N) availability is a major limiting factor for crop growth and productivity in most agricultural production systems [128,129]. The symbiotic relationship between the host plant and rhizosphere microbes forms nodules in the root that enable nitrogen fixation in legumes [130]. The host plant adjusts this symbiotic relationship in response to N deficiency. Researchers observed a negative correlation between soil N availability and the microRNA (miRNA) miR2111 abundance in *Lotus japonicus* [131]. The shoot produces miR2111, which is transferred to the root to trigger TOO MUCH LOVE (TML) to control the formation of nodules. Changes in soil N content would trigger changes in the levels of miR2111 to control nodulation [132]. Some beneficial microbes trigger another form of plant immunity called induced systemic resistance (ISR) to protect the host plant against pathogens and herbivore attacks. The transcription factor MYB72 is identified as a key regulator of ISR [133]. Under iron-deficient conditions, MYB72 triggers the production of iron-mobilizing phenolic metabolites that are released into the rhizosphere to increase iron solubility for acquisition and reconfigure rhizosphere microbiota [134].

Several of the molecular mechanisms that govern the aforementioned processes are still not clear and more research is required to reveal host and microbe changes during an active *in planta* interaction [135]. In-vivo biosensors can play a pivotal role in discovering these changes over the entire lifecycle of the plant.

## 3. Existing Optical Sensing Technologies to Monitor Plant-Microbial Interactions

Studying the interplay between microbes and plants would allow an understanding of how microbes govern crop growth. To better assess soil microbial interactions and eventually plant health, it is crucial to monitor the type and quantity of bacteria and their growth in real time. Plant root exudates are the primary communication system between plants and microbes dwelling in the rhizosphere. Initially, root exudates were analyzed using standard chemistry methods. Since the 1950s, paper chromatography and 14 C isotope labeling have been used for the separation and detection of both sugars and amino acids [136]. Later, various techniques have been applied for structure-based analysis and identification of root exudate compounds. These techniques include Nuclear Magnetic Resonance (NMR), Gas Chromatography-Mass Spectrometry (GC-MS), Liquid Chromatography (LC), and Capillary electrophoresis (CE) [137]. The conventional GC-MS, LC, and NMR imaging are discrete, disruptive, in vitro, and time- and labor-intensive. In addition, because of adsorption on the matrix, microbial degradation, and root damage/stress during sampling, the results are less reproducible [138]. The infrared and thermal imaging techniques for in situ monitoring rely on indirect quantification techniques, lack accuracy, and do not provide quantitative analysis of biomolecules [139]. Remote sensing techniques using unmanned aerial systems do not provide chemical profiling of the plant and are very power-hungry, frequently requiring human intervention for battery recharge/replacement [140,141,142]. There are no commercial in-situ plant sensors (e.g., the stem sensors developed by FloraPulse, Dynamax, and PlantDitech lack chemical profiling) available to provide real-time and continuous monitoring of biomolecule dynamics in plants. Researchers have developed different techniques to monitor the plant-microbe interactions over time. For instance, a microfluidic chip combined with microscopy is developed for continuous observation of interactions between live roots and rhizobacteria in *Populus tremuloides* over 5 weeks [143]. However, such bulky setups are limited to laboratory analysis and not suitable for conducting studies at the field scale. Biosensors are expected to have competitive advantages over the existing methods by providing continuous in-situ monitoring capabilities, multiplexed detection of biochemicals, wireless data transfer capability, energy-efficient and low-cost solutions, and robustness to environmental variations.

Over the past few decades, several biosensing techniques have been developed for monitoring microbial activity in various soil environments, and to unravel how hosts and microbes interact at the gene expression level [144]. To develop effective and just-in-time disease management strategies and understand environmental impacts on plant fitness in real-time, plant-pathogen interactions need to be studied by field-deployable, low-cost, handheld biosensors or genomics and metagenomics analysis [145]. Biosensors have emerged as sophisticated detection instruments in a variety of research disciplines, including environmental monitoring, real-time detection of airborne pathogens, and monitoring pesticide residues in food or drink [146]. The majority of these biosensors involve a bioreceptor element that provides recognition specificity through selective biochemical interactions. In presence of the target analyte or pathogen in the medium, the biosensor transduces the biomolecular interactions into a measurable electrical, mechanical, optical, or acoustic signal [147]. Hence, different types of transducers including electrochemical, mechanical, thermal, piezoelectric, or optical can be developed based on specific requirements such as target bacteria/pathogen/molecules, sensitivity, specificity, limit-of-detection, reproducibility, etc. [145]. Due to the non-contact nature of measuring light, optical sensors can overcome many problems typically associated with live-plant studies, such as wounding caused during measurements [148]. Plant surfaces can be profiled by various methods such as Raman spectroscopy, Positron emission tomography, X-ray fluorescence spectroscopy, or spectral imaging that do not require any optical probes [149,150,151]. Although there exist non-optical complementary techniques capable of measuring biomolecular interactions such as protein fragment complementation (PFC) [152], force spectroscopy (FS) [153], molecular recognition imaging (MRI) [154], and dielectrophoretic (DEP) tweezers [155], and optical biosensing probes provide a competitive advantage in terms of in situ, real-time, and low-cost sensing. This review article is centered around the optical biosensing probes, devices, and/or platforms that demonstrate the potential for in-situ plant monitoring at a much lower cost and simpler means.

Optical biosensors have demonstrated efficacy in sensing microbe-specific molecules such as enzymes, antibodies, antigens, receptors, nucleic acids, whole cells, and tissues. The optical biosensor produces a measurable optical signal in response to the interaction between the reporter and the inducer molecules. Biosensors with optical detection capabilities provide real-time and label-free detection of biological and chemical substances at low costs, compared to traditional analytical techniques [156]. For example, optical DNA hybridization biosensors have been developed for easy and quick microbe infection detection in plants. These biosensors provide polymerase chain reaction (PCR)-independent visual inspection of nucleic acid offering simplicity, high sensitivity, and rapid results [157]. This section presents a review of optical biosensors for detecting biomolecules that are directly associated with plant-microbial interactions. This section is subdivided into subsections based on the type of optical sensing technology. The performance metrics of various biosensors are then compared in Table 2 at the end of this section.

### 3.1. Localized Surface Plasmon Resonance Biosensors

Luna-Moreno et al., [158] developed a localized surface plasmon resonance (LSPR)-based immunosensor for detecting *Pseudocercospora (P.) fijiensis,* a fungal pathogen known to cause Black Sigatoka (leaf streak disease) in banana plants. The sensor was comprised of a gold-coated lateral flow assay immobilized with polyclonal antibody molecules that targeted HF1, a cell wall protein of *P. fijiensis* (Figure 3a). The sensor exhibited a linear response in the range from 39.1 to 122 µg mL^−1^ of HF1 and a detection limit of 11.7 µg mL^−1^. The device thus demonstrates a potential for in-field monitoring of Black Sigatoka disease. The *Tomato yellow leaf curl viru*s (TYLCV) genome was detected in infected plants by a DNA-based nano-biosensor using LSPR [159]. A mixture of unmodified gold nanoparticles (AuNPs) and a specific DNA probe complementary to the virus genome was used to develop a colorimetric assay, which could detect the DNA extracted from TYLCV-infected tomato plants. The assay was sensitive to the extracted DNA in a concentration ranging from 0.75 to 200 ng/µL. The presence of TYLCV genome in 5 ng of the extracted DNA was detected without any amplification. This proposed method provided a fast alternative to traditional PCR-based amplification and detection. In another work, *Cucumber green mottle mosaic virus* (CGMMV) was detected using unmodified gold nanoparticles as colorimetric probes [160]. The underlying working principle relied on the specific binding of CGMMV RNA to the gold nanoparticles thereby enhancing the resistance to NaCl-induced aggregation of the nanoparticles. As a result, the attachment of the RNA probes to the gold nanoparticles led to a color change from red to blue. A concentration of as low as 30 pg/µL of CGMMV RNA was detected through the proposed colorimetric assay. This method was the first demonstration of unmodified gold nanoparticles as colorimetric probes for CGMMV detection in real samples.

Attenuated total reflection (ATR)-based sensors utilize evanescent wave absorption techniques for real-time detection of biomolecules [161,162,163,164]. In this regard, an ATR platform was modified to LSPR to detect the single-stranded DNA (ssDNA) of the chili leaf curl virus (ChiLCV; Genus: *Begomovirus*, Family: Geminiviridae) [165]. The amine-functionalized surface was immobilized with gold nanoparticles to generate LSPR. Figure 3b shows the dynamic variations in absorbance due to specific (viral ssDNA) and non-specific (healthy ssDNA) binding to the LSPR probes. The proposed ATR-LSPR platform exhibited a detection limit of 1 µg/mL for the ChiLCV DNA. Lavanya and Arun [166] reported a colorimetric LSPR assay for early detection of Begomovirus in chili and tomato plants Total DNA was isolated from the plant samples and subsequently analyzed by the LSPR assay, as schematically illustrated in Figure 3c. The assay was able to detect up to 500 ag/µL of begomoviral DNA. (pTZCCPp3, a clone carrying partial coat protein gene).

Fiber-optic biosensors have garnered much attention owing to their high aspect ratio footprint, remote sensing capability, and ease of multiplexing [167,168,169,170,171,172,173]. Our research group developed a gold nanoparticle-coated optical fiber and demonstrated its application as an LSPR sensor for plant salicylic acid (SA) detection [20]. To realize the LSPR sensor, the distal end of an optical fiber was coated with a conjugate of gold nanoparticles and copper-based metal-organic framework (CuMOF) for selective detection of SA. The fiber-tip sensor was inserted into the stem of a live cabbage plant via a micron-size hole (Figure 3d). The light spectrum reflected from the sensor tip had LSPR dips at 542, 676, and 730 nm, as shown in Figure 3e. A nearly linear shift in LSPR intensity was observed in response to increasing SA concentrations. The fiber-tip LSPR sensor demonstrated a linear SA detection range from 100–1000 μM and a detection limit of 37 μM.

Molecularly imprinted polymer (MIP)-coated LSPR sensors show promise in detecting VOCs released by plants in response to pathogenic diseases and pest damage. Shang et al. [174] developed an LSPR sensor by doping gold nanoparticles into molecularly imprinted sol-gel. The gold nanoparticles substantially increase the signal intensity by generating hot spots. An array of sensors was used to identify four plant VOCs, cis-jasmone, α-pinene, limonene, and γ-terpinene, and their mixtures. A pattern recognition algorithm was used to differentiate the clusters of VOC samples. Analysis based on the K-nearest neighbor showed that VOCs were identified with 96.03% accuracy. A handheld smartphone-based VOC fingerprinting device was developed for diagnosing late blight caused by *Phytophthora infestans* [175]. The device incorporated a sensor array that was comprised of cysteine-functionalized plasmonic nanoparticles and chemo-responsive organic dyes. The platform could detect sub-ppm levels of (E)-2-hexenal (a primary VOC released during *P. infestans* infection), with a detection accuracy of >95%.

### 3.2. Lateral Flow Immunoassays

Lateral flow immunoassays (LFIAs) are yet another popular technique for the on-site detection of a wide variety of compounds including pathogens [176]. These are immunochromatographic assays comprised of metal nanoparticles, antibodies against the pathogen, and signal amplification probes functionalized in a test strip. The working principle of LFIAs is shown in Figure 4. Microbial diseases account for 20–40% of losses in crop yield every year [177]. In tropical and subtropical plants, most of the bacteria are beneficial or saprotrophic, which entails that they do not cause any damage to the cell. However, there are approximately 100 bacterial species that can cause infection including gall, overgrowth, blight, leaf spot, and soft rot [156].

Razo et al. [178] reported a sensitive LFIA by enlarging gold nanoparticles’ size to detect *Ralstonia solanacearum,* which causes potato brown rot. *The gold enhancement approach resulted in a lower detection limit of* 3 × 10^4^ cells/mL (Figure 5a) in the potato tuber extract and a combined assay preparation and detection time of only 15 min. The improved performance could be attributed to the catalytic properties of gold nanoparticles. Silver enhancement can be used to detect potato leafroll virus (PLRV) with high sensitivity [179]. A substantial signal enhancement was observed owing to the reduction in silver ions on the surface of gold nanoparticles. The enhancement took place due to the addition of a drop of silver lactate and hydroquinone mixture. Figure 5b shows a comparative demonstration of LFIA in conventional sandwich format (i) versus silver-enhanced sandwich format (ii). 

### 3.3. Lux Bioluminescent Biosensors

Bioluminescence is a non-invasive technique for in-situ sensing and analysis in a variety of applications [180]. Lux biosensors are living bacteria cells that contain a hybrid plasmid with two basic elements, namely a regulatory region (promoter, operator) and a reporter gene [181]. The lux genes isolated from bacteria are used as reporter genes in these biosensors [182]. Lux biosensors are suitable for long-term monitoring of a variety of sugars, polyols, organic acids, amino acids, and flavonoids present in root exudates. The presence of these substances was validated using a metabolomics technique, which could identify 376 molecules in pea root exudate [183]. The symbiosis between pea (*Pisum sativum*) plants and the bacterium *Rhizobium leguminosarum bv. viciae* is a well-established model system for understanding plant-microbe interactions (and has been successfully used to investigate how root exudates affect bacterial gene expression) [184]. The interaction between the root exudates and the bacteria living at the root nodules can be analyzed in real time by monitoring sugars, polyols, organic acids, and amino acids in a non-destructive semi-quantitative manner [183]. For example, a suite of Lux bioreporters was developed with *Rhizobium leguminosarum bv. viciae* strain 3841 to examine the root exudates and rhizobial colonization in plant roots [183]. Pea roots were imaged every 3 to 4 days until 22 days of post inoculation. This long-term monitoring revealed changes in the microbiota and its interplay with root exudates. The strongest *Lux* signals were obtained from the root elongation zone where numerous metabolites are exuded, while comparatively lower Lux signals were observed at the root cap due to the reduced colonization in this zone. The compounds that govern different stages of rhizobium-legume association were identified. Sugars and polyols (sucrose, fructose, erythritol, mannitol, and myo-inositol), organic acids (formate, malonate, tartrate, C4-dicarboxylate, and salicylic acid), and amino acids (flavonoids and hesperetin) were detected. Dicarboxylates and sucrose were found to be primary carbon sources within the nodules, whereas high levels of myo-inositol were observed prior to nodule formation. Among the amino acid biosensors, the γ-aminobutyrate biosensor was active only inside the nodules, while the phenylalanine bioreporter exhibited a high signal in the rhizosphere. These results demonstrate the potential application of these lux bioreporters in unraveling the dynamics of root exudation and how that is engineered by the rhizosphere. In another study, bacterial biosensors were used to analyze the relationship between shoot nitrate concentration and root exudation from Hordeum vulgare [185]. A bacterial biosensor (*Pseudomonas fluorescens* 10586 pUCD607) tagged with the *lux* CDABE genes was developed in this regard. This study was carried out over a 28-day period. Plants were first grown in C-free sand microcosms for 14 days and were supplied with 2 mM nitrate solution. In the next 14 days, the plants were subjected to three treatments: (a) continuous supply with 2 mM nitrate, (b) increase the nitrate application to 10 mM, and (c) further boost the nitrate application to 20 mM. At the end of the treatment period, the lux biosensor was used to measure the C-substrate availability resulting from the root exudation. Imaging of biosensor bioluminescence revealed that decreasing shoot nitrate concentration resulted in increased root exudation. This was attributed to the systemic plant responses to internal N-deficiency.

Luminescence-based bacterial biosensors are also used in detecting secondary metabolites secreted by plant roots in the rhizosphere. Plants and bacteria can synthesize a wide variety of metabolic compounds to sustain their cells in harsh environments. For example, under salt and water stress conditions, the production of proline, which is highly water solubilized and a scavenger of reactive oxygen species, has been found to possess protective benefits [186]. Proline exudation shows chemotactic effects in alfalfa roots and serves as a source of energy, carbon, and nitrogen in environmental stress conditions [187,188]. Proline concentrations in pea root exudates under water-stress conditions were detected using a lux biosensor. The biosensor was constructed by cloning the promoter sequence of pRL120553 [189].

### 3.4. Fluorescence Resonance Energy Transfer Biosensors

Fluorescence resonance energy transfer (FRET) is another promising technology for measuring metabolite levels and their rate changes in living cells. FRET refers to the nonradiative transfer of energy from a donor fluorophore to an acceptor fluorophore through intermolecular dipole-dipole coupling [190]. FRET energy transfer is highly efficient when the donor and acceptor molecules are separated by a distance equivalent to the Förster radius (typically 3–6 nm). Fluorimetry assays have been successfully applied to mammalian and plant cells, but could also potentially be used to monitor steady-state metabolite levels in microorganisms [191]. In addition, these biosensors can detect ligands in vitro as well as in vivo and monitor metabolites in various cells and cellular compartments in real time [192]. Additionally, the FRET technique can be ratiometric, allowing for quantitative and calibrated recordings. Due to the constitutive expression of the biosensor, a signal can be recorded immediately [193]. FRET biosensors have been used to detect twenty-two compounds, divided into eight classes including two hexoses, two pentoses, two disaccharides, two nucleotides, six ions, four amino acids, one nucleobase, and three phytoestrogens [192]. Plant phytohormones regulate various stages of plant growth and metabolism and are indicators of environmental stresses. Several FRET biosensors have been reported to detect these phytohormones. For example, ABA (Abscisic acid), a phytohormone primarily linked to heat and drought stress, was measured in Arabidopsis plants in vivo with a FRET probe. The probe was constructed with an ABA-specific optogenetic reporter linked with a green fluorescent protein. The FRET reporter was called ‘ABAleon’, which was composed of the protein pair mTurquoise- cpVenus173 [194]. Figure 6a shows the structural configuration of ABAleon. In the absence of ABA, there was FRET from mTurquoise (mT) to cpVenus173 (cpV173). In contrast, the presence of ABA led to the formation of PYR1 (Pyrabactin resistance 1)-_ΔN_ABI1 complex, which increased the distance between the fluorescent probes, thus reducing FRET efficiency (Figure 6b). It was observed that ABAleon showed a faster response to 10 µM ABA in the pyl4ple mutant as compared to the Arabidopsis Columbia-0 wild type (Figure 6c–f). ABAleon was capable of mapping ABA concentration changes in plant tissues with high spatial and temporal resolution. Another type of ABA reporter with complementary biochemical properties such as “ABACUS” was also reported in the literature [195]. The combined properties of ABAleon and ABACUS could be utilized to study novel ABA signaling in plants. Similarly, plants use auxins to regulate a variety of processes including growth as well as environmental cues. [196] developed a FRET probe termed “AuxSens” for direct visualization of IAA, the most abundant auxin in plants. Toward this end, a dexamethasone-inducible AuxSens construct was introduced in Arabidopsis. The reported AuxSens biosensor directly determined auxin gradients and distribution during the lifespan of the plant. This tool can be very useful in monitoring the spatiotemporal dynamics of auxin in plants.

### 3.5. Fluorometric Biosensors

Plant roots exude a substantial number of carbon-containing compounds that include soluble carbohydrates, such as sucrose, fructose, and glucose. These carbohydrates are an important element of root exudate, and hence it is important to quantify their movements from roots into the rhizosphere. A gel-based enzyme-coupled colorimetric and fluorometric biosensor (comprised of horse radish peroxidase, glucose oxidase, and Ampliflu Red coating) was developed for in-vivo imaging of the spatiotemporal variations in glucose released from the roots of soybeans, cotton, sorghum, wheat, and rice seedlings [197]. When the assay was exposed to glucose, the glucose oxidase in the assay converted glucose to gluconolactone and hydrogen peroxide. Following this reaction, the horse radish peroxidase catalyzed the reaction between the released hydrogen peroxide and the nonfluorescent Ampliflu Red, to form magenta-colored (and fluorescent) resorufin. The primary roots of maize (*Zea mays*) released more glucose from the root base rather than from the root tip, while a reduction in the glucose level was observed under water stress conditions. This study also revealed the differential spatial variability in glucose exudation in different plant species. Biosensors based on fluorometric techniques can reveal fascinating details about the physiology of free-living soil bacteria. 

Ref. [198] engineered a strain of the bacterium *Sinorhizobium meliloti* that contained a gfp gene fused to the melA promoter. The fusion strain worked as a biosensor, which was activated by galactose and galactosides. This fluorometric biosensor was used to measure the release of galactosides from legume seeds during germination. The biosensor was capable of detecting the presence of galactosides on and around roots in unsterilized soil as well as the grazing of fluorescent bacteria. Lew et al. introduced optical nanosensors to monitor post-wounding H_2_O_2_ profile in lettuce (*Lactuca sativa*), arugula (*Eruca sativa*), spinach (*Spinacia oleracea*), strawberry blite (*Blitum capitatum*), sorrel (*Rumex acetosa*) and *Arabidopsis thaliana* [199]. H_2_O_2_ is the primary reactive oxygen molecule that mediates rapid systemic signaling in plants in response to any kind of mechanical wound or injury. The nanosensor was made of DNA functionalized single-walled carbon nanotube (SWCNT) exhibiting fluorescence in the near infrared range. For in-planta sensing, the SWCNT nanoparticles were injected into the plant. Upon binding with the endogenous H_2_O_2_, the fluorescence intensity of DNA-wrapped SWCNT quenched that was monitored under laser excitation (785 nm, 10 mW). This plant nanobionic approach would help monitor plant physiology in real-time and with minimal damage.

**Table 2 micromachines-14-00195-t002:** Performance comparison of optical biosensing technologies for detecting plant microbial responses.

Ref.	Target	Sensor Configuration	Detection Technique	Sensitivity	Analyte Concentration Range/Limit of Detection (LOD)	Major Advantages and/or Limitations
[20]	Salicylic acid (SA)	Conjugate of gold nanoparticles and copper-based metal-organic framework	Fiber-tip LSPR	0.0117% light reflection per μM	Conc. range: 100–1000 μM LOD: 37 μM	+ in situ and real-time detection directly in plant sap + involves minimal damage to the plant + no need to extract plant samples + provides quantitative measurement + rapid detection in 1–2 min - the optical source and detector are bulky and not chip-scale, preventing scalability
[158]	cell wall protein of *Pseudocercospora fijiensis*	Gold-coated lateral flow assay immobilized with polyclonal antibody targeting a cell wall protein of *P. fijiensis*	LSPR	0.0021 units of reflectance per ng mL^−1^	Conc. range: 39.1 to 122 µg mL^−1^ LOD: 11.7 µg mL^−1^	+ reusable platform for routine monitoring + no matrix effects are observed during the sensor performance using real leaf banana extracts. - need to extract plant samples, which incurs a time delay between sample collection and analysis - destructive sample collection
[159]	*Tomato yellow leaf curl viru*s (TYLCV) genome	Unmodified gold nanoparticles mixed with a complementary DNA probe	LSPR colorimetry	-	Conc. range: 0.75 to 200 ng/µL LOD: genome detection in 5 ng of the extracted DNA	+ fast and sensitive detection, eliminating the need for sophisticated PCR amplification and detection equipment - extracted DNA sample goes through multiple steps: mixing with the designed probe, denaturing, annealing, and then cooling to room temperature followed by AuNPs addition. - lacks quantitative measurement - DNA sample needs to be extracted from infected leaves
[160]	*Cucumber green mottle mosaic virus* (CGMMV) RNA	Unmodified gold nanoparticles mixed with a species-specific probe	LSPR colorimetry	-	LOD: 30 pg/µL	+ simple, low-cost, and visual detection + eliminates the need for sophisticated, expensive instrumentation + 100% specificity with good reproducibility - lacks quantitative measurement - RNA sample needs to be extracted from infected leaves and fruits
[165]	Single-stranded DNA (ssDNA) of the chili leaf curl virus (ChiLCV)	Amine-functionalized surface immobilized with gold nanoparticles and complementary ssDNA	ATR-LSPR	0.833 a.u./(µg/mL)	Conc. range: 0.5 to 3.5 µg/mL LOD: 1 µg/mL	+ provides quantitative measurement + the setup was capable to measure binding kinetics - the Kretschmann prism configuration resulted in a bulky and complex optical setup - sample needs to be extracted from infected plants
[166]	Begomovirus DNA	Functionalized gold nanoparticles	LSPR colorimetry	-	Conc. range: 1 ng/µL to 1 ag/µL LOD: 500 ag/µL	+ the detection efficiency of LSPR assay (77.7%) was found to be better than PCR screening (49.4%) + able to detect begomoviruses infecting plants belonging to different genera + five different probes were designed to detect any differences in the detection limit or specificity among the probes - The DNA extraction procedure is lengthy and requires technical expertise
[174]	cis-jasmone, α-pinene, limonene, and γ-terpinene VOCs	Gold nanoparticles doped into molecularly imprinted sol-gel	LSPR	-	LOD: vapor flow rate of 0.3 L/min	+ enhanced sensitivity through hot spot generation + detect plant VOCs in single and binary mixtures using a multichannel sensor configuration + sensing combined with a pattern recognition approach to establish plant VOC identification models. - additional setup is required to generate plant VOC vapor using the headspace method, which is not scalable for in-field monitoring - the sensor is not suitable for in-planta VOC detection
[175]	(E)-2-hexenal VOC emitted during *Phytophthora infestans* infection	sensor array comprised of cysteine-functionalized plasmonic nanoparticles and chemo-responsive organic dyes	LSPR	-	LOD: between 2.5 and 5 ppm	+ smartphone-based handheld VOC fingerprinting platform + in-field monitoring + detects key plant volatiles at the ppm level within 1 min of reaction + early detection of tomato late blight 2 days after inoculation + a detection accuracy of ≥95% - lacks automation in monitoring. The user has to screen every plant manually with the handheld device
[178]	*Ralstonia solanacearum*	Gold nanoparticles functionalized with antibodies	LFIA	-	Conc. range: 10^1^–10^8^ cells/mL LOD: 3 × 10^4^ cells/mL	+ signal enhancement reduced the detection limit by 33 times + rapid detection in 3 min + quantitative assay - requires sample extraction, which is destructive and hinders real-time sensing - sensor was tested with artificially contaminated samples
[179]	Potato leafroll virus	Sandwich of gold nanoparticles and silver enhancement	LFIA	-	Conc. Range: 0.1–100 ng/mL LOD: 0.2 ng/mL	+ silver enhancement makes the assay 15 times more sensitive + up to 0.2 ng/mL of PLRV can be detected with the naked eye - leaves are crushed in a mortar for sample collection - specificity reduces for non-specific virus concentration >1000 ng/mL
[183]	Root exudates	*Rhizobium leguminosarum bv. viciae* strain 3841	Lux	-	LOD: 0.001 mM for sugars and polyols, 0.01 mM for organic acids, and 0.001 mM for amino acids	+ in-vivo spatial and temporal mapping of 376 molecules in pea root exudate + non-destructive sensing - no analysis is presented on the toxic effect of the lux-marked biosensors on the plant or soil - the stress response of the plant upon injection of foreign objects was also not analyzed
[185]	C-substrate availability	*Pseudomonas fluorescens* 10586 pUCD607) tagged with the *lux* CDABE	Lux	-	-	+ determine the relationship between shoot nitrate concentration and root exudation in vivo - lacks information on sensitivity, detection limit, and stability of the sensor
[194]	ABA	FRET reporter termed ABAleon	FRET	-	Conc. range: 0.8–50 μM LOD: ~0.8 μM	+ direct visualization of ABA concentration changes and distribution - binding to the reporter may reduce the amount of ABA that is available to perform its role as a hormone
[196]	IAA	FRET reporter termed AuxSens	FRET	0.8 (FRET ratio)/(10^−2^–10^−4^) M IAA	LOD: ~1 μM	+ quantitative in-vivo visualization of auxin distribution in plants - further analysis of the stability and toxicity of the reporter is needed
[197]	Glucose	A compound of horse radish peroxidase, glucose oxidase, and Ampliflu Red	Fluorescence	-	LOD: down to 7 ng min^−1^ root^−1^ was shown	+ detects spatial variability of glucose released from plant roots - the roots were placed separately on a gel
[198]	Galactosides	A strain of the bacterium *Sinorhizobium meliloti* containing a gfp gene fused to the melA promoter	Fluorescence	-	-	+ non-destructive method to examine rhizosphere soil chemical composition - lacks information on sensitivity, detection limit, and stability of the sensor
[199]	H_2_O_2_ profile	DNA functionalized SWCNT	Fluorescence	-	Conc. Range: 10^−8^–10^−1^ M	+ species-independent nanosensor probe + a simulation model explains the differences in H_2_O_2_ wave velocity across species - requires a non-portable optical setup comprising a laser, camera lens, and filter wheel

## 4. Other Biosensors

Besides optical detection probes, electrochemistry is another popular biosensing technique that several researchers have adopted to study plant-microbial interactions. A detailed investigation of existing electrochemical sensors is out of the scope of this review paper. Hence, a brief overview of some recently reported electrochemical sensors targeted toward measuring plant signaling is provided below.

An immunosensor was developed by immobilizing gold nanoparticles on a porous graphene layer-coated glassy carbon electrode to detect indole-3-acetic acid (IAA) with a detection limit down to 0.016 ng/mL [200]. The sensor also demonstrated selective identification of IAA in the extracted plant seed samples. A three-electrode-based electrochemical biosensor was developed for real-time monitoring of the expression of beta-glucuronidase in tobacco leaves under heat shock [201]. The sensor demonstrated a sensitivity of 0.076 mA/mM-cm^2^ and a limit of detection of 0.1 mM. The enzyme activity was detected 12–26 h after applying the heat shock. Due to the instable nature of IAA and SA, it is difficult to detect these two phytochemicals using high-performance liquid chromatography (HPLC) or mass spectrometry in a timely manner. In this regard, Ref. [202] developed a paper-based electrochemical sensor by modifying conductive carbon tape electrodes with carbon nanotubes to measure IAA and SA at the level of ng. This method allowed rapid and low-cost detection of IAA and SA in tiny plant samples. Electrochemical DNA biosensors have been widely used for the detection of pathogenic diseases in plants. Researchers have reported several DNA-based electrochemical biosensors using redox active components, enzymes, and nanoparticles labeled onto PCR products for improved electrochemical detection [203]. Lau et al. [204] developed a highly sensitive method for plant pathogen detection by combining RPA (recombinase polymerase amplification) with a gold nanoparticles-based electrochemical biosensor. The assay could identify *P. syringae* infection in *A. thaliana* before the appearance of symptoms. Ref. [205] used AHK4 CHASE (cyclases/histidine kinases associated with sensory extracellular domain) to construct an electrochemical cytokinin biosensor with ferrocene as the electrochemical mediator. Upon binding of cytokinin with AHK4, electron transfer between ferrocene and the electrode was hindered, resulting in a decrease in the ferrocene redox peak current. The biosensor has a linear detection range from 50–400 nM and a detection limit of 1.5 nM. The sensor exhibited a selective response to cytokinins in bean sprouts.

Despite demonstrating excellent performance, the aforementioned biosensors suffer from several bottlenecks. These sensors lack spatial and temporal monitoring because of the inability to install them on the plant root, shoot, or leaves. The invasive sample collection procedure prior to analysis results in mechanical wounding and incurs additional stress on the plant. In contrast, in-situ plant monitoring requires a flexible platform that is wearable and does not incur any damage to the plant. However, wearable biosensing for plants is a heavily underexplored area. Our group routinely works on research involving screen-printed and microneedle-structured electrochemical sensors that are wearable to the plant and provide in situ and continuous measurements of plant health. We have developed a multiplexed, bioagent-free, three-electrode-based electrochemical sensor, which was screen-printed on a flexible polyimide substrate to measure the salicylic acid (SA) levels directly in plant leaves [22]. The working electrode was modified with a composite of copper-based metal-organic framework that selectively oxidized SA. In another work, we reconfigured the hormone sensor by incorporating 3D-printed microneedles, which detected hormone levels in the leaf sap [206]. Each microneedle-structured electrode contained an array of microneedles with a height of ~800 μm. The sensor was installed on the leaves to detect SA and IAA levels with a relative error of <1% (when compared with high-performance liquid chromatography measurements). A detection limit down to 0.10 μM was achieved. Moreover, temperature correction of the measured hormone levels was achieved with an in-built temperature sensor. In addition, two sensors were mounted at different heights (0.5 and 6.5 cm) of the same plant. The sensors could accurately measure the SA dynamics across the plant, as was evident from the difference in the SA rise time (3 h) captured by the leaf sensors. We also developed a sensor for stem to monitor sap SA and pH levels in response to water stress conditions [21]. The sensor was very small, like a beetle that plants can easily wear without any discomfort. The sensors were demonstrated to measure SA and IAA levels in the leaves and stems of a live plant under water stress conditions. A noticeable time-series correlation was observed between the hormone levels and the water stress periods. These sensors were capable of identifying the stress level from Day 1. Besides investigating the role of phytohormones in plants subjected to water stress conditions, we also studied the sap pH level variations under salinity stress using a leaf-scale microneedle sensor [19]. We observed that with increasing salt (i.e., NaCl) concentrations in the soil, sap pH decreased in response to the antioxidative defense response triggered in the plant. Recently, we reported a hybrid sensor suite that contained microneedle-structured electrodes for measuring SA and IAA, as well as electrodes printed onto a flexible sheet for monitoring ethylene emitted from fruits [23]. A small drone was used to deploy the sensor suite to fruits before they were harvested. The dynamic measurements of SA, IAA, and ethylene informed the fruit ripeness, which can play a pivotal role in avoiding untimely harvest. All these wearable biosensing platforms are readily reconfigurable to accommodate electrodes for other analytes of interest.

## 5. Conclusions and Future Prospects

This article reviews dynamic mechanisms and biomolecules underlying plant-microbial interactions. The microbial community associated with plant roots is largely influenced by soil. In addition, plants play a significant role in shaping the microbiome taxonomy. Root exudates serve as signaling messengers that enable the communication between soil microbes and plant roots, thereby promoting microbial activity around roots and eventually nutrient uptake by the plant. The shoot is also a rich source of phytohormones that work as the plant’s first responders to environmental stresses. Therefore, tracking these signaling molecules would provide real-time interpretation of root-microbiome interactions under various environmental conditions. Despite the plenty of studies on processes that govern communication of plants with microbial communities, very less is known about the specific strains of microorganisms that provide nutrition to host plants or about how rhizospheric microbiome composition affects nutrient availability. In order to engineer complex microbial consortia with predictable behavior and robust outcomes, we must gain a better understanding of the dynamic interactions of plants with the surrounding microbiota and environment during their entire lifecycle. This article also provides a detailed overview of existing optical biosensing technologies reported in the literature for investigating plant/microbial interactions. These biosensors have versatile applications in monitoring plant physiology and disease progressions. Biosensors with optical transduction capabilities provide real-time and label-free detection of biological and chemical substances at low costs and point-of-care, compared to traditional analytical techniques. As such this paper should be of great interest to readers and/or researchers in the fields of plant and soil sciences as well as biosensors by providing an how the existing sensing platforms can be improved to to engineer a new photonic system. For example, according to the Consultative Group on International Agricultural Research, the Phaseolus Vulgaris, aka commons beans, are a significant source of protein, carbohydrates, vitamins, and minerals for more than 300 million people in the tropics. A novel photonic device or technique that allows us to better understand the interactions of the Phaseolus Vulgaris plant with the surrounding microbiota and environment during its lifecycle is one of the perspectives that this manuscript could offer to the scientific community. Further advances in sensing technology are needed for the early diagnosis of plant diseases and nutrient deficiency. With advanced sensing, it will also become possible to deploy engineered microbiomes safely and effectively in large-scale field settings to substantially improve plant growth. Furthermore, microorganisms can be used for bioelectricity generation from the biomass and biological wastes [207]. This bioelectricity can in turn power the biosensors deployed in the field, thereby resulting in self-powered sensors.

## Figures and Tables

**Figure 1 micromachines-14-00195-f001:**
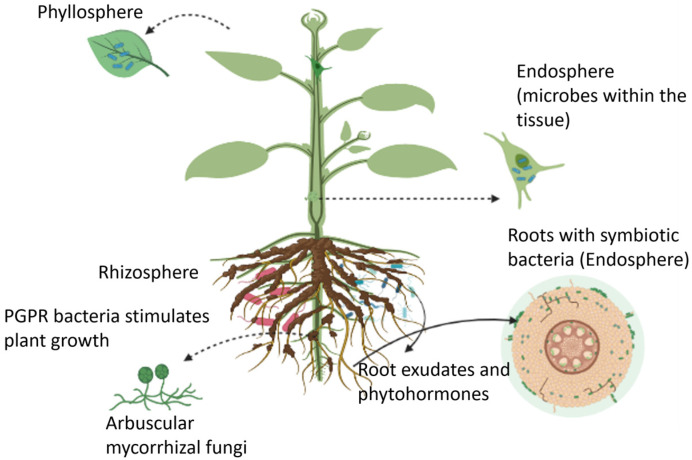
An overview of plant growth-promoting microbes that reside in the rhizosphere, endosphere, and phyllosphere regions.

**Figure 2 micromachines-14-00195-f002:**
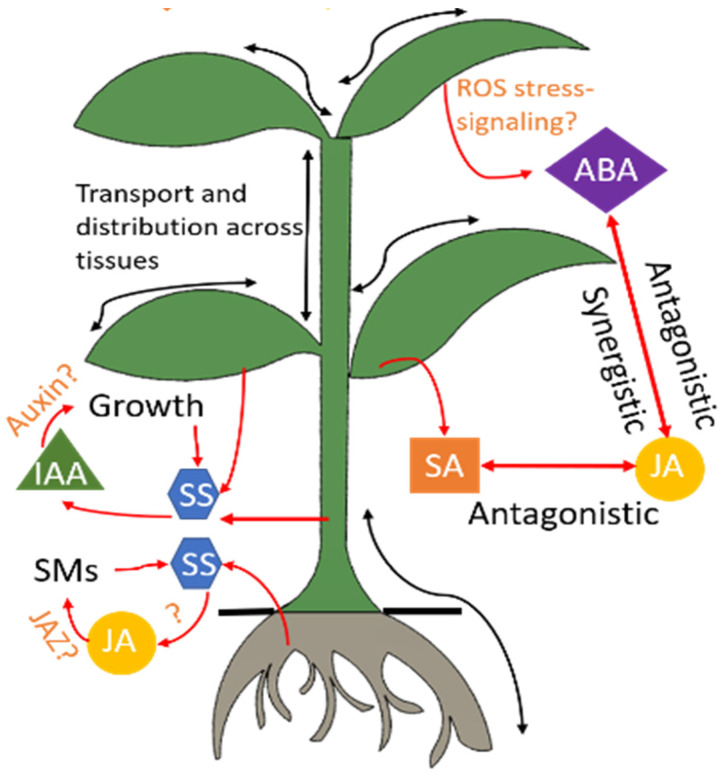
Hormonal regulation in the plant. Stress conditions trigger a cascade of phytohormones. Several signaling pathways of these hormones are still under research and hence indicated by the symbol “?.” Advanced sensors will advance understanding of these interactions through in-situ and real-time monitoring of these hormones. ABA: abscisic acid, IAA: indole-3-acetic acid, JA: jasmonic acid, JAZ: jasmonate, ROS: reactive oxygen species, SA: salicylic acid, SM: secondary metabolites, SS: soluble sugars.

**Figure 3 micromachines-14-00195-f003:**
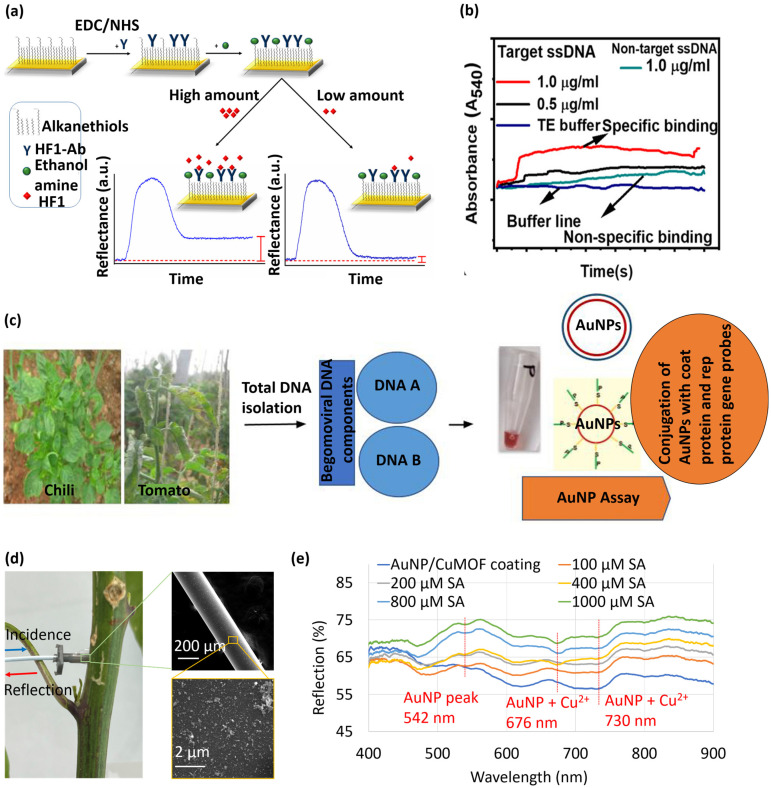
(**a**) Sensograms demonstrating the detection of HF1 by the LSPR immunoassay [158]. Here, EDC = 1-ethyl-3-(3-dimethylaminopropyl) carbodiimide and NHS = N-hydroxysuccinimide. Copyright © MDPI, Luna-Moreno et al. (**b**) Dynamic variation in absorbance due to specific (viral ssDNA) and non-specific (healthy ssDNA) binding to the LSPR probes [165]. Copyright © ACS, Das et al. (**c**) Process flow for detecting Begomovirus in chili and tomato plants using functionalized gold nanoparticles [166] copyright © Springer Nature, Lavanya et al. (http://creativecommons.org/licenses/by/4.0/, accessed on 29 December 2022) (**d**,**e**) in situ SA measurements in the stem of a live plant and resulting reflection spectra for SA sensing in plant sap [20] Copyright © SPIE, Tabassum.

**Figure 4 micromachines-14-00195-f004:**
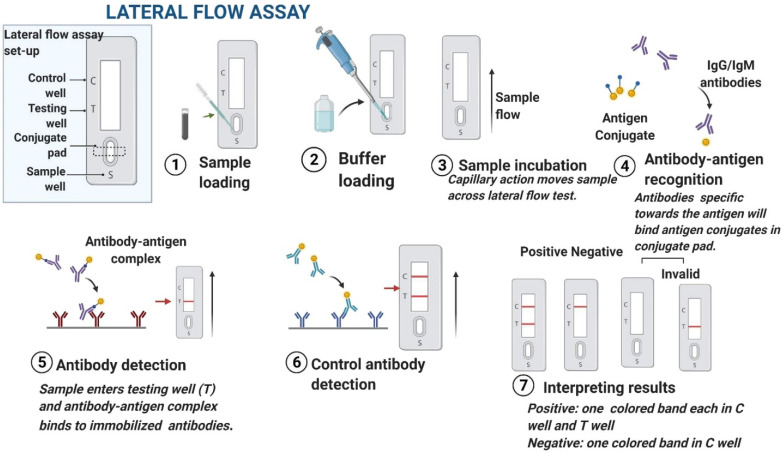
Process flow of an LFIA [176] Copyright © Elsevier, Patel et al.

**Figure 5 micromachines-14-00195-f005:**
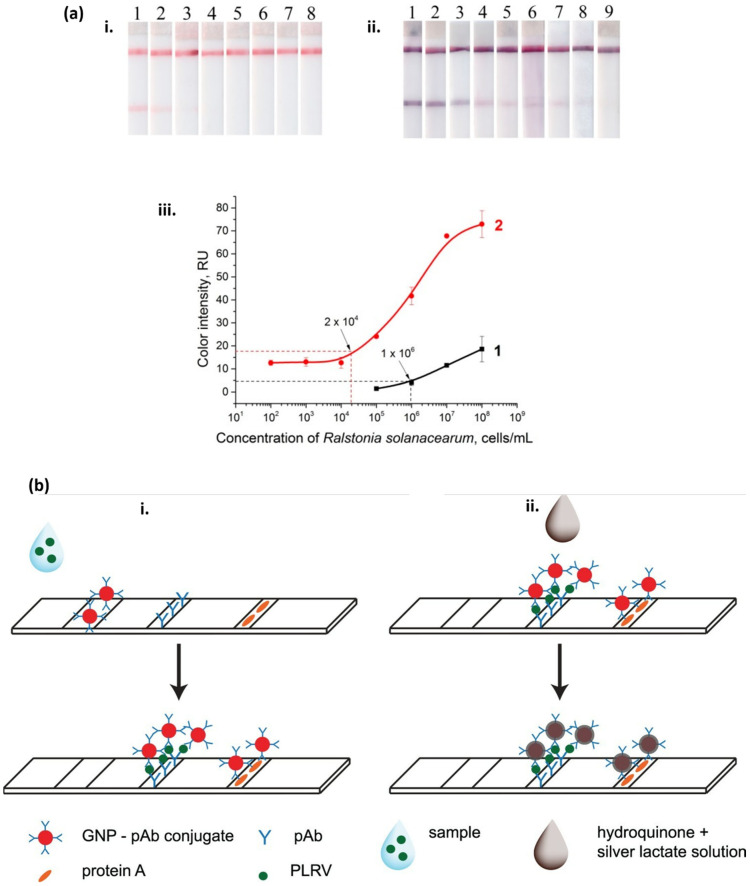
(**a**) Testing of LFIA (**i**) before and (**ii**) after enlargement. Here, strips #1–9 correspond to 10^8^, 10^7^, 10^6^, 10^5^, 10^4^, 10^3^, 10^2^, 10^1^, and 0 cells/mL of *Ralstonia solanacearum, respectively*. (**iii**) calibration curves before (=1) and after (=2) enlargement [178]. Copyright © MDPI, Razo et al. (https://creativecommons.org/licenses/by/4.0/, accessed on 29 December 2022) (**b**) Schemes of LFIAs in (**i**) conventional sandwich format and (**ii**) silver-enhanced sandwich format. Here, GNP = gold nanoparticles, pAb = anti-PLRV antibodies, and PLRV = potato leafroll virus [179]. Copyright © Taylor & Francis, Panferov et al.

**Figure 6 micromachines-14-00195-f006:**
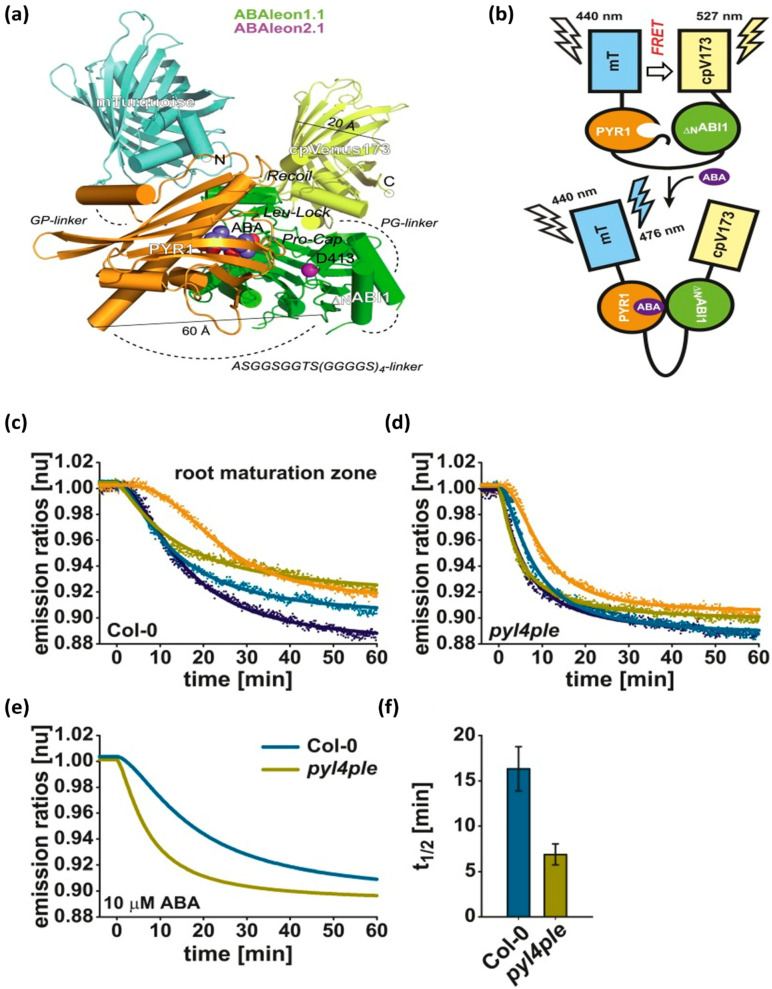
(**a**) Structural features of ABAleon, where mTurquoise (mT) (cyan color) is fused to PYR1 (gold color) and _ΔN_ABI1 (green color) is fused to cpVenus173 (cpV173) (yellow color). (**b**) FRET from mT to cpV173 without ABA and FRET quenching with ABA. ABA response curves of (**c**) Columbia-0 wild type and (**d**) pyl4ple. (**e**) Data from four single measurements were fitted by a four-parameter logistic curve to obtain the combined dataset. (**f**) The t_1/2_ values calculated from the fitted curves in (**c**,**d**) [194] Copyright © eLifeSciences, Waadt et al.

**Table 1 micromachines-14-00195-t001:** List of microbes that produce phytohormones.

Microbial Strains	Phytohormone/Root Exudate	Host Plant/Source	Reference
*Pseudomonas fluorescens* CHA0, WCS374, WCS417, Pf4–92, *P. aeruginosa* 7NSK2, *Serratia marcescens*, P. fluorescens Pf4–92	SA	Tobacco, potato, wheat, cucumber, barley, and chickpeas	[73]
*Ralstonia solanacearum*	Ethylene	Banana	[77]
*P. fluorescens* SPB2145, PCL1751, *Stenotrophomonas rhizophila* e-p10	IAA	Cucumber plant	[90]
*Bradyrhizobium* sp. *Azospirillum* sp. *Bacillus pumilus* and *Bacillus licheniformis*	Gibberellin	*Phaseolus lunatus*, *Alnus glutinosa*, and L. *Pinus pinea* plants	[91]
*Azospirillum lipoferum*, *Arthrobacter koreensis*, *Achromobacter xylosoxidans*, *Bacillus licheniformis*, *Bacillus pumilus*, and *Brevibacterium halotolerans*	ABA	helianthus annuus and rhizobacteria (PGPR)	[92]

## Data Availability

Not applicable.

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
