# Peer review of "Optical Sensing Technologies to Elucidate the Interplay between Plant and Microbes"

_micromachines, 2023, doi:10.3390/mi14010195_

Round 1

Reviewer 1 Report

Comments to the manuscript submitted by Asia Neelam and Shawana Tabassum

The submitted manuscript is a review entitled “Optical Sensing Technologies to Elucidate the Interplay between Plants and Microbes”.

I wish to start by saying that this review possesses relevant content to the biosensors’ field, particularly for the agriphotonics field, due to the optical techniques applied to the in-vivo and in-situ experiments.

The authors show a broad and in-depth knowledge of plant-microbe interactions, allowing a good text organization and a good reading experience. In addition, the references used to elaborate the text sequence are adequate, contributing to easy reading.

Table 2 is well presented with enough references used as examples giving the reader a general idea of the type of optical detection techniques that have already been tested to target different compounds.

That being said, this manuscript is suitable to be published as a review because it can give an understanding of what could be done in the future to engineer a new photonic system (or improve existing ones). For example, the Phaseolus Vulgaris, aka commons beans, are, according to the Consultative Group on International Agricultural Research, provides protein, complex carbohydrates, and valuable micronutrients for more than 300 million people in the tropics. A novel photonic device or technique that allows us to better understand the interactions of the Phaseolus Vulgaris plant with the surrounding microbiota and environment during its lifecycle is one of the perspectives that this manuscript could offer to the scientific community.

Author Response

We thank the reviewer for providing valuable insights into the manuscript. Particularly, we are grateful for the reviewer's comment on Phaseolus Vulgaris. Hence, in the conclusion section, we have added and highlighted a couple of sentences following the reviewer's comments. 

Reviewer 2 Report

The article reviews various optical biosensing technologies that have been developed for investigating plant-microbe interactions and notes the potential for such technologies to be used for monitoring plant physiology and disease progression, as well as for deploying engineered microbiomes in field settings to improve plant growth. The manuscript is of potential interest to the broad readers of Micromachines.

However, there are some problems, which must be solved before it is considered for publication. If the following problems are well-addressed, this reviewer believes that the essential contribution of this paper is important for Plant-microbe interactions.

1. Consideration of environmental factors: In order to gain a better understanding of how plant-microbe interactions are influenced by the environment, researchers could study these interactions under a range of different conditions, including different soil types, temperature regimes, and levels of water availability.

2. Long-term monitoring: To better understand the dynamics of plant-microbe interactions over time, researchers could conduct long-term studies that track changes in the microbiome and related processes over the course of a plant's life cycle.

3. The format for the journal name, year, and volume in the references should be consistent. Careful proofreading is needed.

Author Response

We thank the reviewer for the insightful comments. All the comments are addressed in the revised manuscript and the changes are highlighted. Please see below a point-by-point response to the reviewer's comments. 

  1. Consideration of environmental factors: In order to gain a better understanding of how plant-microbe interactions are influenced by the environment, researchers could study these interactions under a range of different conditions, including different soil types, temperature regimes, and levels of water availability.

Author Response: We thank the reviewer for this insightful comment. In this regard, we have added a separate section titled '2.4 Influence of Environment on Plant-Microbe Interactions' in line 237. The entire section is highlighted in the revised manuscript.

  1. Long-term monitoring: To better understand the dynamics of plant-microbe interactions over time, researchers could conduct long-term studies that track changes in the microbiome and related processes over the course of a plant's life cycle.

Author Response: Thank you very much for this comment. Actually, the lux biosensors that are explained in the manuscript were designed for long-term monitoring of plant-microbe interactions. To clarify this further, we have added additional texts in lines 424-425, 432-433, 445-449, and 302-306. 

  1. The format for the journal name, year, and volume in the references should be consistent. Careful proofreading is needed.

Author Response: Thank you for pointing this out. We have corrected all the references. 

Reviewer 3 Report

The manuscript titled “Optical Sensing Technologies to Elucidate the Interplay between Plant and Microbes” by Neelam, A.; et al. is a review work where the authors present the most recent examples where disruptive optical sensing techniques are used to decipher the interactions taken place between plants and microbes. This is an extensive work which can significantly contribute for the examined field. The knowledge acquired in the present work could significantly aid to better understand the importance of plant-microbe interctions at different levels which can have strong implications in biosphere and Industry. The cases shown are illustrative and well-discussed during the main body of the reported manuscript. The scientific paper is well written. In my opinion the present manuscript is innovative and the methodological approached used matches with the scope of Micromachines journal. For the above described reasons, I recommend the publication in Micromachines once the following remarks will be fixed:

--------

1. INTRODUCTION

The Introduction is clear and concise. Nevertheless, some minor remarks should be covered in order to improve the scientific quality of the manuscript.

“Moreover, understanding the dynamics of and crosstalk between the hormonal signaling pathways would elucidate the defense mechanisms in plants” (lines 55-57). Please, the authors should correct the English typo “of an” of this sentence. Additionally, it may be convenient to add a relevant reference at the end of this statement [1].

[1] Bhar, A.; et al. Plant Responses to Biotic Stress: Old Memories Matter. Plants 2021, 11, 84. https://doi.org/10.3390/plants11010084.

“This article is organized into several sections. Section 1 provides (…)” (line 77). (OPTIONAL) I may change “Section 1” by “This section” or “The current/present section”.

--------

2. THE INTERPLAY BETWEEN PLANTS AND MICROBIAL COMMUNITIES

“(…) Pseudomonas syringae (P. syringae) (…)” (lines 127-128). Please, the authors should highlight the microorganism “Pesudomonas syringae” in Italics.

“Plants also release 100 Tg of methanol (…)” (line 154). Please, the authors should define the term “Tg” by adding teragram. Then, the abbreviation should appear between brackets.

--------

3. EXISTING OPTICAL SENSING TECHNOLOGIES TO MONITOR PLANT-MICROBIAL INTERACTIONS

“the stem sensors developed by (…) PlantDitech, (…)” (line 325). Why did the authors underline the name of the company PlantDitech?

“Due to the non-contact nature of measuring light, optical sensors can overcome (…) optical probes” (lines 346-350). Here, even if I agree with the authors, it may be relevant to name non-optical complementary techniques capable to measure biomolecular interactions like protein fragment complementation (PFC) [2], force spectroscopy (FS) [3], molecular recognition imaging (MRI) [4] and dielectrophoretic (DEP) tweezers [5]. Then, the authors perfectly explain the main advantages presented by optical methods compared to these aforementioned techniques.

[2] Singh, A.; et al. Dissecting virulence pathways of Mycobacterium tuberculosis through protein-protein association. Proc. Natl. Acad. Sci. U.S.A. 2006, 103, 11346-11351. https://doi.org/10.1073/pnas.0602817103.

[3] Pérez-Domínguez, S.; et al. Nanomechanical Study of Enzyme: Coenzyme Complexes: Bipartite Sites in Plastidic Ferredoxin NADP+ Reductase for the Interaction with NADP. Antioxidants 2022, 11, 537. https://doi.org/10.3390/antiox11030537.

[4] Marcuello, C.; et al. Molecular Recognition of Proteins through Quantitative Force Maps at Single Molecule Level. Biomolecules 2022, 12, 594. https://doi.org/10.3390/biom12040594.

[5] Kim, M.H.; et al. Automated Dielectrophoretic Tweezers-Based Force Spectroscopy System in a Microfluidic Device. Sensors 2017, 17, 2272. https://doi.org/10.3390/s17102272.

Figure 3 (line 427). Could it possible to enlarge the font size of the information detailed in the panel (a)? This fact will significantly aid the readability to the potential readers. Additionally, the abbreviations “EDC” ((1-ethyl-3-(3-dimethylaminopropyl)carbodiimide) and “NHS” (N-hydroxysuccinimide) should be defined in the respective place of this Figure caption.

Table 2 (line 588). The authors should add two columns to briefly indicate the main advantages and limitations of each optical biosensing technique in comparison to the rest.

--------

4. OTHER BIOSENSORS

The authors perfectly state the most relevant details about alternative biosensors. No further actions are requested for this section.

--------

5. CONCLUSION AND FUTURE PROSPECTS

This section is well structured. The authors perfectly remark the most relevant insights provided in this work. The authors depict some potential future avenues in the application of optical biosensing technologies “Further advances in sensing technology are needed for the early diagnosis of plant diseases and nutrient deficiency” (lines 682-683). Nevertheless, it may be convenient to showcase some potential Industrial applications like the design of good practices to improve the plant development for the production of food, key metabolites or biofuels [6].

[6] Kumar, R.; Kumar, P. Future Microbial Applications for Bioenergy Production: A Perspective. Front. Microbiol. 2017, 8, 450. https://doi.org/10.3389/fmicb.2017.00450.

--------

REFERENCES

Bibliography citations are not in the proper format of I Micromachines journal. The authors should address this point.

--------

OVERVIEW AND FINAL COMMENTS

The submitted work is well-designed and the displayed information is interesting to summarize the most recent examples in the use of optical biosensing devices to study the intermolecular interactions of plants and microbes. For this reason, I will recommend the present scientific manuscript for further publication in Micromachines once all the aforementioned suggestions will be properly fixed.

Author Response

We thank the reviewer for the insightful comments. All the comments are addressed in the revised manuscript and the changes are highlighted. Please see below a point-by-point response to the reviewer's comments. 

  1. INTRODUCTION

The Introduction is clear and concise. Nevertheless, some minor remarks should be covered in order to improve the scientific quality of the manuscript.

“Moreover, understanding the dynamics of and crosstalk between the hormonal signaling pathways would elucidate the defense mechanisms in plants” (lines 55-57). Please, the authors should correct the English typo “of an” of this sentence. Additionally, it may be convenient to add a relevant reference at the end of this statement [1].

[1] Bhar, A.; et al. Plant Responses to Biotic Stress: Old Memories Matter. Plants 202111, 84. https://doi.org/10.3390/plants11010084.

“This article is organized into several sections. Section 1 provides (…)” (line 77). (OPTIONAL) I may change “Section 1” by “This section” or “The current/present section”.

Author Response: Thank you very much for this comment. We have corrected the aforementioned items in lines 50-52 and 67 of the revised manuscript. 

  1. THE INTERPLAY BETWEEN PLANTS AND MICROBIAL COMMUNITIES

“(…) Pseudomonas syringae (P. syringae) (…)” (lines 127-128). Please, the authors should highlight the microorganism “Pesudomonas syringae” in Italics.

“Plants also release 100 Tg of methanol (…)” (line 154). Please, the authors should define the term “Tg” by adding teragram. Then, the abbreviation should appear between brackets.

Author Response: Thank you very much for catching the mistake. We have corrected this. Please see lines 105 and 123-124 of the revised manuscript. 

  1. EXISTING OPTICAL SENSING TECHNOLOGIES TO MONITOR PLANT-MICROBIAL INTERACTIONS

“the stem sensors developed by (…) PlantDitech, (…)” (line 325). Why did the authors underline the name of the company PlantDitech?

“Due to the non-contact nature of measuring light, optical sensors can overcome (…) optical probes” (lines 346-350). Here, even if I agree with the authors, it may be relevant to name non-optical complementary techniques capable to measure biomolecular interactions like protein fragment complementation (PFC) [2], force spectroscopy (FS) [3], molecular recognition imaging (MRI) [4] and dielectrophoretic (DEP) tweezers [5]. Then, the authors perfectly explain the main advantages presented by optical methods compared to these aforementioned techniques.

[2] Singh, A.; et al. Dissecting virulence pathways of Mycobacterium tuberculosis through protein-protein association. Proc. Natl. Acad. Sci. U.S.A2006103, 11346-11351. https://doi.org/10.1073/pnas.0602817103.

[3] Pérez-Domínguez, S.; et al. Nanomechanical Study of Enzyme: Coenzyme Complexes: Bipartite Sites in Plastidic Ferredoxin NADP+ Reductase for the Interaction with NADP. Antioxidants 202211, 537. https://doi.org/10.3390/antiox11030537.

[4] Marcuello, C.; et al. Molecular Recognition of Proteins through Quantitative Force Maps at Single Molecule Level. Biomolecules 202212, 594. https://doi.org/10.3390/biom12040594.

[5] Kim, M.H.; et al. Automated Dielectrophoretic Tweezers-Based Force Spectroscopy System in a Microfluidic Device. Sensors 201717, 2272. https://doi.org/10.3390/s17102272.

Figure 3 (line 427). Could it possible to enlarge the font size of the information detailed in the panel (a)? This fact will significantly aid the readability to the potential readers. Additionally, the abbreviations “EDC” ((1-ethyl-3-(3-dimethylaminopropyl)carbodiimide) and “NHS” (N-hydroxysuccinimide) should be defined in the respective place of this Figure caption.

Table 2 (line 588). The authors should add two columns to briefly indicate the main advantages and limitations of each optical biosensing technique in comparison to the rest.

Author Response: We thank the reviewer for the insightful comments. 

We have removed the underline from PlantDitech and line 301.

 As per the reviewer's suggestion, we have explained non-optical complementary techniques capable to measure biomolecular interactions in lines 324-327. 

We have enlarged the font size in Figure 3a and added the full forms for EDC and NHS in the figure caption. Please see lines 386-388. 

We have corrected Table 2 (line 519) and added one column to briefly highlight the main advantages and limitations of each optical biosensing technique. 

  1. OTHER BIOSENSORS

The authors perfectly state the most relevant details about alternative biosensors. No further actions are requested for this section.

Author Response: We thank the reviewer for pointing this out. 

  1. CONCLUSION AND FUTURE PROSPECTS

This section is well structured. The authors perfectly remark the most relevant insights provided in this work. The authors depict some potential future avenues in the application of optical biosensing technologies “Further advances in sensing technology are needed for the early diagnosis of plant diseases and nutrient deficiency” (lines 682-683). Nevertheless, it may be convenient to showcase some potential Industrial applications like the design of good practices to improve the plant development for the production of food, key metabolites or biofuels [6].

[6] Kumar, R.; Kumar, P. Future Microbial Applications for Bioenergy Production: A Perspective. Front. Microbiol20178, 450. https://doi.org/10.3389/fmicb.2017.00450.

Author Response: We thank the reviewer for the insightful comment. We have added a brief discussion as highlighted in lines 600-602.

REFERENCES

Bibliography citations are not in the proper format of I Micromachines journal. The authors should address this point.

Author Response: Thank you for pointing this out. We have corrected all the references.